# Ciliary Function, Antigen Stasis and Asthma

**DOI:** 10.3390/ijms251810043

**Published:** 2024-09-18

**Authors:** Nadzeya Marozkina

**Affiliations:** School of Medicine, Indiana University, Indianapolis, IN 46256, USA; nmarozki@iu.edu; Tel.: +1-317-274-7427

**Keywords:** primary ciliary dyskinesia, asthma, nitric oxide, oxidative stress

## Abstract

The prevalence of asthma exceeds 3% of the population. Asthma is observed to be more common in children following severe viral lower respiratory illnesses that affect ciliary function, but mechanisms linking ciliary function to asthma pathogenesis have been obscure. Recent data regarding primary ciliary dyskinesia (PCD) may help us to understand the association. Here, I will review what is known about the relationship between ciliary function and asthma. PCD is caused by pathologic variants in over 50 different genes that affect the structure and function of motile cilia. At the cellular level, a characteristic feature shared by most PCD patients is that antigens and other particles are not cleared from the epithelial surface. Poor antigen clearance results in pro-oxidant pathway activation and airway epithelial damage and may predispose PCD patients to DUOX1- and IL33-mediated asthma. Secondary ciliary dysfunction, such as that caused by viruses or by smoking, can also contribute to asthma development. Moreover, variants in genes that affect the function of cilia can be associated with poor lung function, even in the absence of PCD, and with increased asthma severity. The role of antigen stasis on the surface of dysfunctional airway cilia in the pathophysiology of asthma is a novel area for research, because specific airway clearance techniques and other therapeutic interventions, such as antioxidants, could be of value in preventing the development of asthma.

## 1. Introduction


**Antigen Stasis in Cilia as Predisposition for Asthma.**


Asthma is a chronic condition affecting over 3% of the world’s population [1,2,3,4]. It is defined as reversible airflow obstruction with underlying airway inflammation. Typically, chronic airway exposure to antigens and/or irritants leads to airway swelling and constriction of the smooth muscle outside the airway. This can be mediated by several cytokines, chemokines, alarmins and neural signaling molecules. Classically, these include T helper type 2 (Th2) mediators such as interleukin (IL) 4, IL5 and IL13, or non-Th2 cytokines such as IL-6, IL-17, IF-γ and TNF-α. Upstream of both Th2 and non-Th2 airway inflammation is IL33 (as well as other molecules) that is secreted in response to airway injury by epithelial cells. Recent evidence suggests that decreased ciliary function can lead to the stasis of antigens on the epithelial surface [5]. This leads to airway inflammation and helps explain the clinical association between asthma and both primary and secondary causes of ciliary dysfunction [1,6,7,8]. It is these associations that are the subject of this review.

First, it is important to explain that primary ciliary dyskinesia (PCD) is an inherited condition that is caused by pathologic variants in over 50 different genes that affect the structure, repair and function of motile cilia [9]. PCD is more common than was previously thought, with a prevalence of about half that of cystic fibrosis: at least 1 in 7600 people [10]. The diagnosis of PCD is challenging, and the measurement of nasal NO (nNO) has been used as a biomarker for diagnosis in patients with PCD symptoms. These clinical attributes include newborn respiratory distress; chronic nasal congestion; daily cough, bronchitis, and bronchiectasis; and disorders of left–right symmetry (both situs inversus totalis and heterotaxy) [6,9,10,11,12,13]. Most patients with a confirmed diagnosis of PCD have low nNO [6,11,12]; however, the reason for this is unknown.

At the cellular level, a characteristic feature shared by most PCD patients is that antigens and other particles are not well mobilized from the epithelial surface [14]. We have shown that PCD cells have decreased clearance of the antigen Dermatophagoides pteronyssinus (Derp-1) from the cell surface compared to normal airway epithelia [5]. Derp-1 is a house dust mite (HDM) allergen that belongs to the group-1 allergens, which commonly induce allergies. We and others have shown that antigen stasis on the airway epithelial surface results in the upregulation of pro-oxidant enzymes that promote airway inflammation [15,16,17,18,19]. It is important to note here that these studies have been carried out with primary cell cultures grown at the air–liquid interface from healthy control subjects and PCD patients. This is the most carefully controlled method for studying the redox biochemistry of airway epithelial cells: direct sampling of airway epithelial cells from the nose can give rise to a number of artifacts, making the redox data difficult to interpret. Here, the thesis of this review is that antigen stasis, particularly in children, leads to airway inflammation and to asthma-like symptoms in patients with PCD.

Our evidence suggests that PCD can cause airway injury and low airway levels of NO through the upregulation of oxidant enzymes in the epithelium. In PCD, nNO levels are so low that measurement has become a component of the consensus ATS approach to PCD diagnosis [11]. Expression of NO synthase isoforms is not generally decreased in in PCD airways [20,21,22]. Our data suggest that nNO levels are low because NO is oxidized before leaving the airways. Antigens and irritants stuck on the epithelial surface upregulate the activity of pro-oxidant epithelial enzymes that produce O_2_-, and NO is depleted by forming cytotoxic ONOO-/ONOOH. Additional NO oxidation reactions can deplete NO and form NO_2_, HNO_2_/NO_2_- [23]. These reactions cause airway injury and can lead to bronchospasm [24]. 

Additionally, antigen stasis upregulates DUOX1 [15,16,25], injuring the airway through the formation of H_2_O_2_. One pro-oxidant enzyme is dual oxidase 1 (DUOX1), the upregulation of which during antigen stasis is mediated by P2Y and PAR receptors, leading to the activation of type 2 alarmins [16]. Duox1 upregulation has been directly associated with ciliary movement [25]. P2Y and PAR activation have been linked with the activation of DUOX1, but not necessarily with the upregulation of DUOX1 protein levels. Oxidative stress, in turn, decreases gas-phase NO [5]. While airway NO concentrations are low in children with PCD [9,11,12,26], our evidence demonstrates that the oxidation products of NO in the PCD airway are normal or high [5]. In addition to depleting NO and injuring the airway, each of these reactions have the potential to cause bronchoconstriction and inflammation [7,24,25,27]. Further, when antigen stasis leads to DUOX1 upregulation, there is epithelial release of the local alarmin, interleukin (IL)33 [28]. Downstream, IL33 can interact with its receptor (ST2) to initiate a cascade of mediators leading to asthma, including T-helper 2 (T2) cell-mediated asthma, non-T2 asthma and mixed asthma [8]. Our data to date suggest that PCD-associated asthma will be characterized primarily by non-T2, low FeNO asthmatic airway inflammation, though it could also cause a so-called “mixed” asthma pathophysiology. These possibilities will require further study. Either way, I will argue that PCD, causing antigen stasis, leads commonly to bronchial hyper-reactivity in patients with PCD. Several early publications showed that FeNO levels are low in PCD patients [29,30], arguing that NO from the intrathoracic airway is also decreased. My lab is currently comparing antigen stasis and the NO oxidation response between nasal and bronchial epithelia. So far, there is no difference. My group is also studying whether those PCD patients with asthma have T2, non-T2 or mixed asthma. So far, we have shown that the mediator from epithelial cells is primarily IL-33, which can lead to both. But it is too early to speculate. It will be fascinating to sort this out as the field of PCD-related asthma moves forward. 

## 2. The Majority of PCD Patients May Have Asthma

Many children with PCD require asthma treatment, such as inhaled long-acting β2 agonists (LABAs) and inhaled corticosteroids (ICSs), to prevent episodes of acute bronchoconstriction. Indeed, a recent survey-based review of PCD management in the 22 PCD centers in Italy revealed that all centers prescribed ICSs for their patients [31]. Some centers prescribed them to all patients, while others only if there was a positive test for bronchial hyper-reactivity (BHR), a sine qua non for asthma diagnosis [32]. Two studies have already studied BHR retrospectively, and a formal, prospective study is currently under way. An Israeli retrospective review demonstrated that 56% of PCD patients studied had significant reversal of obstruction following albuterol (BHR) [33] (Table 1). A second Canadian retrospective study of serial clinic visits found that most of their PCD patients had an improvement in forced expiratory volume at 1 s (FEV1) with the β2 agonist, albuterol, but only 18% had a 12% improvement [34] (Table 1). There are several weaknesses in these retrospective studies. Neither study used the accepted methods for analyzing bronchial hyper-responsiveness that have been adopted by NIH programs such as the Severe Asthma research Program (SARP) [1] or the Precision Interventions for Severe and Exacerbation Prone Asthma (PrecISE) networks [2]. For the diagnosis of asthma, large networks like the SARPPrecISE use the ATS standards that first require the subject to stop chronic controller medicines like ICSs and LABAs for at least a day before coming in for BHR testing. Second, a subject can sequentially receive increasing doses of albuterol until there is no further bronchodilatation. Third, if a subject does not fully bronchodilate with albuterol, another drug withhold is carried out before a second visit, and the subject undergoes a methacholine challenge test. If a subject is determined to have BHR by either test (bronchodilatation with albuterol or bronchoconstriction with methacholine—or other bronchoconstricting agents), a diagnosis of asthma is given. None of these requirements were met in either of the retrospective studies. Further, in the Canadian study, the older criterion of 12% improvement in FEV1 was used; the Israeli study used the more recently accepted norm, 10% [34]. Nevertheless, strikingly, most of the PCD subjects in both studies experienced improvement in FEV1 with albuterol. These studies are summarized in Table 1. Note that an additional study, reported in abstract form, is still enrolling; its results will likely be the most conclusive, as it is a prospective study using best-practice guidelines for diagnosis. 

Data recently presented at the annual “PCD On The Move” meeting addressed this issue by two methods. First, large informatic databases were queried to assess whether asthma is common among children with PCD. Though there is *not* an International Classification of Diseases (ICD)10 code for PCD itself, queries were made using a commonly used ICD10 code (Q89.3), as well as established codes for both bronchiectasis and situs inversus totalis, in the Indiana Network for Patient Care (INPCR) electronic health record [HER] database. An additional validation cohort was made using the IBM Explorys database (over 80 million individual EHRs). Patients who had the relevant ICD10 codes were matched to age-, sex- and race-matched controls who did not have those codes at a ratio of 1 PCD subject/3 controls. Informatic analysis showed that subjects who definitively had PCD (both bronchiectasis and situs inversus totalis) had a ~20-fold increased risk of also having an asthma diagnosis. 

Secondly, a prospective study of BHR is ongoing using formal criteria as described above, among our PCD patient population. Approximately 80% of confirmed PCD patients studied to date have a BHR test diagnostic for asthma. These data are consistent with the Israeli study noted above (see Table 1) [33]. Indeed, PCD is often treated like asthma with the routine use of inhaled steroids and β-agonists [35]. All of the above suggests the possibility that the majority of patients with inherited ciliopathies may also have asthma. However, additional studies are needed.

## 3. In Epithelial Cells, Cilia Can Also Serve as Sensory Cilia, and Loss of Epithelia Can Affect Epithelial Signaling and Airway Smooth Muscle Relaxation

As noted above, NO oxidation is likely to be an important reason for the relatively pathognomonic low nNO observed in PCD. This “low-NO” phenotype distinguishes PCD-related asthma from other forms of asthma. Note, however, that there may well be other contributing factors as well, factors that may be of relevance to the relationship between inherited ciliary dysfunction and asthma. First, the decreased expression of NO synthase (NOS) isoforms does not appear to account for the profoundly low nNO levels observed in PCD: expression is not much different in PCD airways compared to healthy individuals [20]. However, decreased NOS *activity* is a possibility, and we have previously reported that decreased ciliary motion prevents airway epithelial endothelial NOS (eNOS) activity [36], and this difference in activity could theoretically contribute to low airway NO values, though there is no difference in expression. A similar effect in endothelial cell signaling has been reported by Straub and coworkers [37], whereby hemoglobin expression and regulation of the heme iron oxidation state determines the bioactivity of colocalized eNOS. Indeed, NOS makes bronchodilator S-nitrosothiols in addition to NO [24,38]. We have shown that hemoglobin (Hb) is present in human airway pseudostratified epithelia [36]. Hbβ mRNA increases during the maturation and ciliation of airway epithelia and Hbβ knockdown decreases ciliation, while overexpression causes loss of ciliation. Hemoglobin in the airway epithelial cell affects nitrogen oxide biology. Apical eNOS, which can signal through passive ciliary motion (airflow) as it does in other ciliated organs [39], is co-localized with Hbβ. Air pressure and flow allow calcium entry into the cell, activating eNOS; eNOS, in turn, produces NO and S-nitrosothiols [36]. CO pre-treatment ablates NOS-dependent [21,24,39] NO_3_^−^ and NO_2_^−^ accumulation in epithelial medium, consistent with its actions in the endothelium. Thus, the Angelo paradigm [40] is relevant in the ciliated epithelium as it is at the myoendothelial junction: eNOS ultimately produces NO and S-nitrosothiols if the heme iron is oxidized, and it produces inert oxidation products if the heme iron is reduced. Airway clearance vest treatment, passively moving airway cilia, increases NOS activity [41]. Thus, ciliary motion increases NOS activity independently of its expression. This is one mechanism by which decreased ciliary function can decrease airway NO levels [36]. 

Note also that ciliary paralysis and dysfunction can affect airway metabolism. Indeed, end-tidal oxygen tension is much higher in patients with PCD than in patients with CF or asthma, likely reflecting decreased effective respiration, and potentially contributing to the oxidative loss of NO, as well as to nitrosative stress [42].

## 4. Secondary Ciliopathy

The findings discussed above suggest that PCD is a risk factor for asthma. Additionally, secondary ciliary dysfunction—such as that caused by common childhood respiratory viruses, likely also contributes to asthma. Respiratory syncytial virus (RSV) is an example. It is a ubiquitous, seasonal respiratory viral pathogen and a major cause of morbidity and mortality in infants (children younger than 12 months) worldwide. Many years of observational studies have consistently shown an association between RSV bronchiolitis and the development of childhood asthma [43,44,45]. A recent meta-analysis has supported this strong association [44]. 

In a recent study of 1946 subjects, healthy children who were not infected with respiratory syncytial virus (RSV) during their first year of life were significantly less likely to have asthma by age of five than uninfected children, according to results from an observational study that included 1946 participants. Thus, preventing RSV infections during infancy could theoretically decrease asthma cases among five-year-olds by up to 15%. Further, the risk of developing asthma was also associated with the severity of their RSV infection: milder RSV infections led to a lower risk of asthma at the age of 5 years than the severe RSV infections did [46].

RSV infects the respiratory epithelium, leading to chronic inflammation [47], and induces the loss of cilia activity, two factors that determine mucus clearance and the increase in sputum volume. Other changes caused by RSV include ultrastructural abnormalities confined to ciliated cells, such as increased cilia loss and mitochondrial damage [47,48]. I hypothesize that secondary ciliary dysmotility caused by viral infections like RSV could, as in PCD, cause antigen stasis, contributing to an asthma diathesis [49]. 

These alterations involve reactive oxygen species-dependent mechanisms. ROS overproduction plays an important role in the pathogenesis of severe RSV infection and serves as a major factor in pulmonary inflammation and tissue damage. Thus, antioxidants seem to be an effective treatment for severe RSV infection [50]. The antioxidant N-acetylcysteine (NAC) has proven useful in the management of COPD, reducing symptoms, exacerbations and accelerated lung function decline in RSV [49]. Antioxidants prescribed for RSV treatment may be useful to manage secondary ciliopathy as well. 

Other pathogens, such as Pseudomonas aeruginosa, can affect respiratory cilia by inhibiting their beating [51], as well as Streptococcus pneumoniae [52] by producing oxidant H_2_O_2_ [25]. Mycotoxins released by opportunistic fungi such as Aspergillus flavus can reduce ciliary motility [53]. The effects of RSV and bacterial airway infections thus can create a vicious cycle. They decrease ciliary function, causing antigen stasis and oxidative stress; this, in turn, worsens ciliary function. 

Toxic chemical exposure, such as that experienced during smoking, can also result in secondary ciliopathies. Smokers have short cilia compared to non-smokers; COPD is associated with further ciliary shortening. [54,55]. Electron microscopy showed that smokers with chronic bronchitis have more ciliary abnormalities than non-smokers; these include compound cilia, giant cilia and other microtubular and axonemal abnormalities [56,57,58]. These abnormalities can persist after smoking cessation [59]. This irritant-induced decrease in ciliary function can lead to antigen stasis, augmenting asthma risk.

## 5. Antigen Stasis Leading to Oxidant Generation Is Increased in the PCD Airway Epithelium and May Contribute to Both Ciliopathies and Asthma Pathogenesis—Antioxidant Treatment May Be Beneficial for PCD Management

Low nasal nitric oxide is now considered to be diagnostic for PCD in children over five years old [12,26,60]. If decreased ciliary function increases oxidative stress in the airway, NO could be consumed, and the airway could be injured. Specifically, NO reacts with oxidants, including H_2_O_2_, O_2_-, OH● and O_2_ itself [24,27,61,62]. Kinetics may have some variables. Products of the reaction include NO_2_, HNO_2_-/NO_2_-, NO_3_-, ONOO-/ONOOH and other nitrogen oxides [24,27,61,62]. All these oxidative reactions deplete the reactant, nitric oxide. In addition to the depletion of NO in the PCD airway, oxidative stress can contribute to airway injury. Superoxide reacts rapidly with NO to form ONOO-/ONOOH (pKa ~6.5), and peroxinitrite reacts with protein tyrosines to cause protein damage through tyrosine nitration [27,62]. Depending on conditions, the products can cause cytotoxicities oxidative stress in general, and products of NO oxidation in particular, which can injure the airway epithelium [24,27,61]. One pro-oxidant enzyme is dual oxidase 1 (DUOX1), the upregulation of which during antigen stasis is mediated by P2Y and PAR receptors, leading to the activation of type 2 alarmins [15,16]. DUOX1 upregulation may be directly associated with cilia movement [25]. Therefore, we hypothesized that PCD airway epithelial cells might not efficiently clear antigens from the surface, resulting in increased oxidative stress. This probability is supported by recent evidence that antigens on the airway epithelial surface increase expression of pro-oxidant enzymes in airway epithelial cells [62,63]. This oxidative stress, in turn, could both decrease gas-phase NO concentrations and contribute to airway nitrosative stress and airway epithelium injury. While it has been shown previously that airway NO concentrations are low in PCD [6,9,11,12,35,60], our evidence demonstrates that the oxidation products of nitric oxide in the PCD airway are normal or high. Obviously, in healthy airways, ciliary function is normal. Antigens and irritants are rapidly cleared, and NO enters the gas phase normally to be exhaled. In the PCD airways, however, we have shown that antigens, such as Dermatophagoides pteronyssinus (Derp)1, and irritants in the airways are not as well cleared [5]. This defect leads to oxidative stress, marked by increased DUOX1 expression and decreased superoxide dismutase [SOD] activity. H_2_O_2_, in high concentrations in the PCD airway, injures the airway. NO is oxidized under these circumstances, primarily by O_2_-, rather than being exhaled. Tissue injury, at least in part through tyrosine nitrosation chemistry, may predispose PCD patients to asthma; additionally, the immune response to antigens that are not cleared from the epithelial surface may promote the asthmatic airway response. 

Thus, we hypothesized that PCD patients may benefit from airway clearance [64] and antioxidant therapy. Antioxidants decrease the production of different oxidants. For example, Apocynin and SOD decrease O_2_-, [65] and, in the presence of NO and ONOOH, catalase decreases levels of H_2_O_2_ [66]. Glutathione and vitamins A, C and E and β-carotenes can have more generalized antioxidant effects [67,68,69]. We have shown that treatment with either apocynin or SOD treatment increases headspace NO over DNAH11 PCD cells [5]. Vitamin D may be beneficial for PCD patients with bronchiectasis and vitamin D deficiency [70]. Mucous retention, antigen stasis and bacterial or viral infection, due to cilia immotility or inefficient motility in PCD patients, promotes infections and inflammation, which in turn can lead to pulmonary exacerbations and bronchiectasis. As a result, quality of life, lung function and structure are relatively poor in PCD patients. Airway clearance (both pharmacological—b-agonists, pulmozymee (DNA-ase), hypertonic saline, antibiotics (tobramycine) and mechanical (Vest, Chest PT)) or physical exercise can be beneficial as well [71]. 

Exhaled NO fraction measurement can be used routinely to evaluate the response and adherence to treatments [72].

## 6. PCD Genes and Asthma Risk

Of the over 50 genes that are known to be causal for PCD, most are autosomal recessive [73]. Some data argue that being a carrier for a recessive PCD gene could be a risk factor for asthma. For example, abnormal expression patterns of RSPH1, RSPH4A, RSPH9 and DNAH5 were significantly more common in patients with nasal polyps, a chronic upper airway inflammatory disease that is frequently triggered by defective host defense [74], even in the absence of classic PCD.

Moving forward, as more genes and variants are discovered, it will be important to continue to track whether impaired ciliary function represented by heterozygosity could represent a risk factor for poor lung function and for asthma. 

A few additional examples suggesting an impact of PCD gene heterozygosity and increased risk for obstructive lung disease are as follows. Variants in genes underlying the development or function of cilia, such as DNAH14 and DNAAF3, were associated with poor lung function in cystic fibrosis, whereas variants in DNAH6 were associated with preserved lung function in cystic fibrosis. Associations between DNAH14 and lung function were replicated in disease-related phenotypes characterized by obstructive lung disease in adults [75]. One study has shown that the expression of PCD-related genes, DNAH5, KIFC2 and KIF3A, are downregulated in asthma, and that SNPs in these genes correlate with asthma and disease severity [76]. Several single-nucleotide polymorphisms (SNPs) in PCDH1 have been linked to asthma and BHR; however, the functional consequences of this mutation have not been explored [77]. Finally, an association of RSPH3 is observed in non-obese children with asthma [69] 

## 7. Conclusions—Asthma Is a Common, Debilitating Condition—Both Inherited and Acquired Ciliopathies Appear to Contribute to Asthma Pathophysiology—Inherited Ciliopathy Leading to Antigen Stasis May Predispose Children to Asthma Development 

Asthma management should be considered in children with PCD. Moreover, a diagnosis of PCD should be considered in some patients with asthma. Basic diagnostic tools such as nasal nitric oxide, high-speed video microscopy analysis (HSVMA), immunofluorescent staining, axonemal ultrastructure analysis via transmission electron microscopy (TEM) and genetic testing are available for MCP/PCD testing. Patients with PCD can certainly be initially diagnosed with asthma. But there are telltale signs of PCD that, if absent, can relatively quickly be reassuring. Centrally, these are the key diagnostic criteria as outlined by Shapiro and others [9,11,12]. Specifically, these include two out of four of the following: perinatal respiratory distress without a clear underlying cause, particularly if associated with atelectasis and prolonged oxygen requirement in a term infant; chronic, daily rhinitis beginning in the first year of life (or chronic sinusitis in an adult); chronic, daily wet cough beginning in the first year of life, or bronchiectasis; and any form of heterotaxy [9,11,12]. If the provider is suspicious that the asthma patient has underlying PCD, the patient can be referred to a PCD Foundation-accredited center for nasal NO testing (this will generally be <77 nL NO/min, and/or a PCD genotype can be sent). Both tests, however, can be falsely negative, and the involvement of a pulmonologist at a PCD center is advisable if there remains lack of clarity. 

Further, strategies should likely be considered to help clear antigens and irritants after acute airway infections such as RSV, for the purpose of preventing antigen stasis-induced asthma. Antigen stasis in PCD upregulates pro-oxidant generation, leading to decreased nNO and airway epithelial injury as well as increased IL33 production. In developing antioxidant treatment regimens, biomarkers such as nNO and breath H_2_O_2_ might be considered as surrogates for the effectiveness of the therapy. Indeed, clinical evidence is beginning to emerge that the majority of PCD patients also have asthma. This association leads to a broader set of questions regarding the development of asthma. Specifically, do secondary ciliopathies predispose individuals to asthma through similar mechanisms; and could partially dysfunctional cilia in some heterozygous patients also be a risk factor for asthma? Either way, the role of cilia and antigen stasis as the pathophysiology of asthma is an important area for research, because specific airway clearance techniques and other therapeutic interventions, such as antioxidants, could be of value in preventing the development of asthma and asthma-like symptoms. This proposed scheme is summarized in Figure 1.

Antigens, such as cockroaches, mites, RSV and chemical irritants such as those produced by smoking, cannot be removed easily from PCD lungs as from healthy lungs due to ciliary immotility. This brings about the activation of pro-oxidant enzymes, such as DUOX1, and increases IL33 production and, ultimately, airway epithelial injury, predisposing PCD lungs toward asthma development. Additional genetic factors, such as genetic variations in some PCD-causing genes (DNAH14, DNAAF3, DNAH6, DNAH5, KIFC2, KIF3A), are associated with the increased risk of asthma development. 

## Figures and Tables

**Figure 1 ijms-25-10043-f001:**
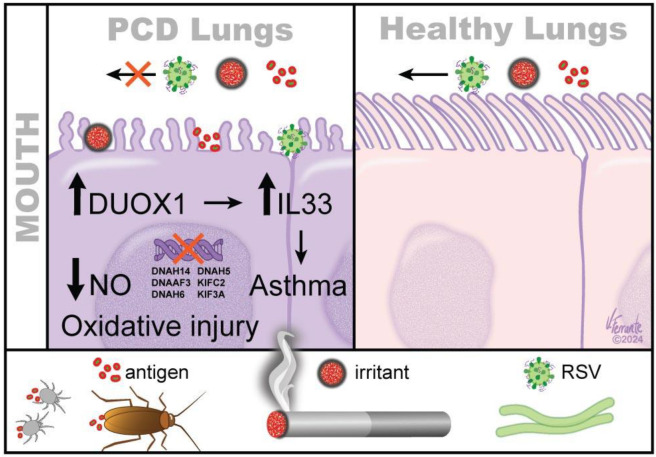
Antigen stasis in PCD increases the risk of asthma.

**Table 1 ijms-25-10043-t001:** PCD patients with positive bronchodilatator response.

Study	Asthma Drug Withhold before the Study	Maximum Bronchodilator	Methacholine If Failed Bronchodilator	Prospective	Number	Percent of PCD Patients with Positive Bronchodilatator Response (BD)	Reference
Indiana	Yes	Yes	Yes	Yes	9	81	
Israel	No	No	No	No	46	56	[33]
Toronto	No	No	No	No	474	18	[34]

## Data Availability

Data sharing is not applicable to this article.

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
