# Peer review of "Ciliary Function, Antigen Stasis and Asthma"

_ijms, 2024, doi:10.3390/ijms251810043_

Round 1

Reviewer 1 Report

Comments and Suggestions for Authors

The article entitled “Ciliary function, antigen stasis and asthma” by Nadzeya Marozkina aims to describe as the title suggest, the ciliary function in asthma. The paper is, however, mostly about primary ciliary dyskinesia and other ciliopathies, not really asthma. The article is not well prepared, it needs extensive corrections not only at the level of the language but also from the editorial point of view. The content of the respective paragraphs should be re-written to a more consistent way and provide only the pivotal information without those irrelevant to the topic. Mental shortcuts and repetitions of information or key words should be avoided (like in titles of paragraphs).

Comments on the Quality of English Language

English language should be corrected.

Author Response

Response: Thank you.  I have made extensive revisions in accordance with these important suggestions.  I think these ideas have significantly improved the readability. Specifically,

  • I have centered the paper on asthma rather than on PCD. I begin with a more thorough description of asthma.  My thesis that ciliary dysfunction can contribute to the development of asthma.  This is still at the heart of the manuscript.  This is the innovative addition to the literature, and it is the reason I think it will be frequently cited.  But I have framed this thesis in a discussion of asthma.
  • I have reviewed the structure to make the manuscript flow better. I believe that you will find it to be less repetitive.
  • I have made use of institutional editorial resources to check the English grammar and usage.

Reviewer 2 Report

Comments and Suggestions for Authors

Marozkina N presents a revision of the literature regarding ciliary function and dysfunction in primary ciliary dyskinesia and asthma and the possible relationship between these two conditions. The major hypothesis supported by the author is that airway stasis of allergens and toxic molecules because of ciliary dysfunction can trigger inflammatory and immunology processes leading to asthma. The review is interesting but some changes are need to increase clearness and readability.

MAJOR COMMENTS:

1.      Introduction has a subtitle (1.1 Antigen stasis in cilia as predisposition for asthma) but no other paragraphs of this section are present. Introduction should be a short section regarding the state of the art about PCD and asthma and then a paragraph about antigen stasis should be the first one

2.      The author frequently reports first person hypothesis and suggestions. These sentences should be deleted (Lines 66-68, lines 96-97, lines 220-221, lines 239-240, etc…). Lines 69-70 is a suggestion not supported by references, the same for lines 93-96 where the author reported personal data without a reference. The same for lines 134-148, personal data not supported by references.

3.      Line 151 Israeli and Italian study? The Italian (or Indiana?) study is not included in the table.

4.      The association between PCD and asthma is not supported in the review by clear data (percentages, frequencies of association…), as well as stasis of antigens as responsible for the association between ciliopathy and asthma

5.      Repetitions are present: Lines 218-219 and 224-225 (RSV includes ultrastructural abnormalities confined to ciliated cells, such as increased cilia loss and mitochondrial damage. RSV includes ultra- structural abnormalities confined to ciliated cells, such as increased cilia loss and mitochondrial damage)

6.      No references have been reported when the author cited the percentages of motile ciliary disorders in patients with bronchiectasis (lines 250-257)

7.      The author reported that blood or systemic inflammation do not affect propensity of MCD (line 259) but in the following sentence she reported that mild MCD was more frequent in patients with bronchiectasis and blood or sputum eosinophilia

8.      The first part of the paragraph 5 Is a repletion of what previously reported

9.      It could be useful to add a brief paragraph showing diagnostic tool for MCD or PCD. Should pneumologist always consider MCD when they evaluate patients with asthma or with severe asthma?

10.   The author tried to explain why NO decreases in the nose of PCD patients. What happens in bronchial epithelial cells? Since asthma, particularly T2 asthma, is characterized by high FeNO levels, could it be that NO is decreased in the nose and not in the bronchial epithelium? Or we can suggest MCD in non T2 asthmatics?

MINOR COMMENTS

1.      Number of bibliography should be formatted (line 235)

2.      Line 250 lack or verb in the sentence

3.      Lines 257-259: The sentence is not clear

4.      The title of Paragraph 3 is not clear and too long, and also the title of paragraph 5 is too long

Comments on the Quality of English Language

English only needs minor editing

Author Response

Reviewer 2.

MAJOR COMMENTS:

  1. Introduction has a subtitle (1.1 Antigen stasis in cilia as predisposition for asthma) but no other paragraphs of this section are present. Introduction should be a short section regarding the state of the art about PCD and asthma and then a paragraph about antigen stasis should be the first one.

Response: Thank you for this important point.  We removed a subtitle 1.1 and made the introduction more uniform.  In response to the first reviewer, we have added more background on asthma to this section.

  1. The author frequently reports first person hypothesis and suggestions. These sentences should be deleted (Lines 66-68, lines 96-97, lines 220-221, lines 239-240, etc…). Lines 69-70 is a suggestion not supported by references, the same for lines 93-96 where the author reported personal data without a reference. The same for lines 134-148, personal data not supported by references.

Response: We have made the following changes in the text, supported by references:

For lines 66-68 I put: Duox1 upregulation has been directly associated with ciliary movement [18].”, “The oxidative stress, in turn, decreases gas phase NO” [7].

For lines 96- 97, I put:  “Neither study used the accepted methods for analyzing bronchial hyper-responsiveness that have been adopted by NIH programs such as the Severe Asthma Research Program [28] or the Precision Interventions for Severe and Exacerbation Prone Asthma (PrecISE) network [29].”

For lines 220-221, I put:  It has been published previously, that secondary ciliary dysmotility caused by viral infections like RSV could, as in PCD, cause antigen stasis, contributing to an asthma diathesis [46].

For lines 239-240, I deleted this sentence.

For lines 69-79, I put:  The oxidative stress, in turn, decreases gas phase NO [7].

For lines 93-96, I have extensively re-written this section for clarity.  The reference to our own work is reference 30. I also changed the heading of section 2 to “The majority of PCD patients may have asthma.”

For lines 135-9, I changed the text to “All above suggest the possibility that the majority of patients with inherited ciliopathies may also have asthma”.

  1. Line 151 Israeli and Italian study? The Italian (or Indiana?) study is not included in the table.

Response:  I’m a bit confused about this question. I don’t see reference to Israel or Italy on line151.  But ref 26 was an Israeli study, and it is included in the table. As far as I have been able to find, there is no Italian study on asthma among PCD patients.

  1. The association between PCD and asthma is not supported in the review by clear data (percentages, frequencies of association…), as well as stasis of antigens as responsible for the association between ciliopathy and asthma. Then point out several places where you have added that additional studies are needed.?

Response: Thank you for this question.  In response, I have provided what I think important clarifications (see pages 3 and 4 of the revised manuscript).  I am providing a point of view for which there is substantial evidence, but for which there still remain hypotheses to be tested. Two groups have published percentages of PCD patients with asthma, and both numbers are over twice the frequency/percent  of the incidence of asthma in the general population.  These are presented in the table.  Our group has additional evidence as well, data that have been presented in abstract form at an international conference and submitted for publication.   I also review that our group and others have shown that there is antigen stasis in the PCD airway [7], and several groups have shown that antigen stasis causes asthmatic inflammation [8-10], in part through IL33.  Thus, what I am saying is not new or unsupported: it is simply “connecting the dots.”   As far as I know, this is the first review to articulate these connections.  As such, I think it will be frequently cited.

The reviewer’s suggestion to point out where additional studies are needed is also excellent.  I have now done this on page 4.

  1. Repetitions are present: Lines 218-219 and 224-225 (RSV includes ultrastructural abnormalities confined to ciliated cells, such as increased cilia loss and mitochondrial damage. RSV includes ultra- structural abnormalities confined to ciliated cells, such as increased cilia loss and mitochondrial damage)

Response: I removed repeated phrase. Thank you.

  1. No references have been reported when the author cited the percentages of motile ciliary disorders in patients with bronchiectasis (lines 250-257)

Response:

Thank you.  References 1,3,4 and 81 show that 61- 98% of PCD patients have bronchiectasis.  I have also added reference 57, looking at the question the other way around:  suggesting that 9% of non-CF bronchiectasis patients have PCD.   

  1. The author reported that blood or systemic inflammation do not affect propensity of MCD (line 259) but in the following sentence she reported that mild MCD was more frequent in patients with bronchiectasis and blood or sputum eosinophilia

Response:

Thank you. I agree that this section was difficult to follow, and I have deleted it.

  1. The first part of the paragraph 5 Is a repletion of what previously reported

Response: I have condensed both sections to avoid repetition.

  1. It could be useful to add a brief paragraph showing diagnostic tool for MCD or PCD. Should pneumologist always consider MCD when they evaluate patients with asthma or with severe asthma?

Response: Yes, the reviewer is correct.  Asthma management should be considered in children with PCD.  Additionally, certain asthma patients should be evaluated for the evidence of PCD if they meet clinical criteria for PCD. I have put a discussion of  put the relevant evaluation criteria and tools on pages 8 and 9.

  1. The author tried to explain why NO decreases in the nose of PCD patients. What happens in bronchial epithelial cells? Since asthma, particularly T2 asthma, is characterized by high FeNO levels, could it be that NO is decreased in the nose and not in the bronchial epithelium? Or we can suggest MCD in non T2 asthmatics?

Response: This is also an important question to clarify.  Several early publications showed that FeNO levels are low in PCD patients (for example, Shoemark A, Wilson R. Bronchial and peripheral airway nitric oxide in primary ciliary dyskinesia and bronchiectasis. Respir Med. 2009;103(5):700-706. doi:10.1016/j.rmed.2008.12.004; and Alsaadi MM, Habib SS, Al Muqhem BA, Aldrees A, Al Zamil JF, Alsadoon HA. Significance of fractional exhaled nitric oxide measurements in detecting primary ciliary dyskinesia in Saudi children. Saudi Med J. 2013;34(1):24-28.), arguing that NO produced by the intrathoracic airway is decreased, as is NO from the nose.  My lab is currently comparing antigen stasis and the NO oxidation response between primary PCD nasal and bronchial epithelial cells in culture.  So far, there is no difference, and we hope a paper will come out soon.  My group is also studying whether those PCD patients with asthma have T2, non-T2 or mixed asthma.  So far, we have shown that the mediator from epithelial cells is primarily IL-33, which can lead to both.  But it is too early to speculate. It will be fascinating to sort this out as the field of PCD-related asthma moves forward.  I have also explained previously that (see ref 17) why high FeNO does not necessarily represent T2 inflammation, and why low FeNO does not necessarily reflect the absence of T2 inflammation.  This is now discussed more on page 2.  These are important studies to do.  The review I have written is, in many ways, simply a road map that investigators can begin to follow to do this work.  I think this will be valuable both for the PCD community and for the asthma community.

MINOR COMMENTS –

  1. Number of bibliography should be formatted (line 235)

Response: Thank you.  This is now corrected.

  1. Line 250 lack or verb in the sentence

Response: Thank you.  I have reviewed the final manuscript to be certain that all relevant parts of speech are in place.

  1. Lines 257-259: The sentence is not clear

Response: I clarified the sentence.

  1. The title of Paragraph 3 is not clear and too long, and also the title of paragraph 5 is too long

Response: I have re-done these paragraphs and their titles.  Thank you. 

Round 2

Reviewer 1 Report

Comments and Suggestions for Authors

The author has made significant efforts to correct the present paper. Still, there is a important difference between the content and style of par. 1, which is still chaotic and 3, which is easy to understand. I would suggest to the author to check the entire paper for spelling mistakes (line 229: “test” not “vest”, lines 298 and 302 “age OF five” etc), to correct forms of chemical structures, to unify the description of types of asthma (Th2 or T2 asthma) and to check and correct the requirement of the Journal for references in the text. Please add in line 32 examples of non-Th2 molecules, in line 33 examples of such molecules and in line 42 units after 7600. Delete the numbers in keywords. Consider to reformulate the entire paragraph in line 103-127 – write it less personal with less references to current unpublished results (if you feel like citing them, add “unpublished results” or so), with less emotions and avoid descriptions like “my lab”, “my group”.

Comments on the Quality of English Language

Check once again spelling. 

Author Response

Response: I would like to thank the reviewer for these helpful suggestions.  I have addressed each of these suggestions and made the relevant changes in the text.  In each case, I have made every effort to avoid the first person singular.

The only thing I have not revised is “vest”: this is a device used clinically for airway clearance; it is actually the correct term!

Reviewer 2 Report

Comments and Suggestions for Authors

The author precisely answers to the raised questions, except for point 3.

As concerns point 3 of may previous revision, the indicated sentence is at lines 197-198 of the revised manuscript and it reports: “These data are consistent with the Israeli and Italian studies noted above (see Table 1) [26]”. So if this is a typo, please amend or explain it better.  

Author Response

Response: Thank you.  We have changed in line 161 of the revised manuscript “These data are consistent with the Israeli study noted above” (see Table 1) [26].  

Round 3

Reviewer 1 Report

Comments and Suggestions for Authors

I am fine with the corrections made by the author. Good luck!